# Akebia Saponin D Inhibits the Inflammatory Reaction by Inhibiting the IL-6-STAT3-DNMT3b Axis and Activating the Nrf2 Pathway

**DOI:** 10.3390/molecules27196236

**Published:** 2022-09-22

**Authors:** Jin-Fang Luo, Hua Zhou, Chon-Kit Lio

**Affiliations:** 1Basic Medical College, Guizhou University of Traditional Chinese Medicine, Guian District, Guiyang 550025, China; 2Guangdong Provincial Hospital of Chinese Medicine, Guangdong Provincial Academy of Chinese Medical Sciences, State Key Laboratory of Dampness Syndrome of Chinese Medicine, Second Affiliated Hospital of Guangzhou University of Chinese Medicine, Guangdong-Hong Kong-Macau Joint Lab on Chinese Medcine and Immune Disease Research, Guangzhou 510006, China; 3Faculty of Chinese Medicine, Macau University of Science and Technology and State Key Laboratory of Quality Research in Chinese Medicine, Macau University of Science and Technology, Taipa, Macao 999078, China

**Keywords:** *Dipsacus asper* Wall. ex Henry, Akbia saponin D, DNMT3b, p-STAT3, Nrf2

## Abstract

Akebia saponin D (ASD) is derived from the *Dipsacus asper* Wall. ex Henry, which is a traditional Chinese medicine commonly used to treat rheumatic arthritis (RA). However, the in-depth mechanism of the anti-inflammatory effect of ASD is still unclear. This study aimed to preliminarily explore the anti-inflammatory effect of ASD and the underlying mechanisms from the perspective of DNA methylation and inflammation-related pathways. We found that ASD significantly reduced the production of multiple inflammatory mediators, including nitric oxide (NO) and prostaglandin E_2_ (PGE_2_), in LPS-induced RAW264.7 cells. The expression of DNA methyltransferase (DNMT) 3b and inducible nitric oxide synthase (iNOS) was also obviously inhibited by the ASD treatment. The protein and mRNA levels of interleukin-6 (IL-6) and tumor necrosis factor-α (TNF-α) were also significantly inhibited by ASD. ASD inhibited the macrophage M1 phenotype, inhibited the high level of DNMT3b, and downregulated the signal transducer and activator of the transcription 3 (STAT3) pathway to exert its anti-inflammatory activity. Furthermore, DNMT3b siRNA and Nrf2 siRNA significantly promoted the anti-inflammatory effect of ASD. Our study demonstrates for the first time that ASD inhibits the IL-6-STAT3-DNMT3b axis and activates the nuclear factor-E2-related factor 2 (Nrf2) signaling pathway to achieve its inhibitory effect on inflammatory reactions.

## 1. Introduction

Recent studies have found that the occurrence and development of inflammation are accompanied by abnormal changes in DNA methylation [1,2]. DNA methylation is the most common epigenetic modification and is mediated by DNA methyltransferases (DNMTs, including DNMT3a and DNMT3b) [3]. Abnormal changes in DNA methylation are also closely related to the development of inflammation-related diseases [4,5]. Among the above methyltransferases, DNMT3b can regulate DNA methylation. DNMT3b also regulates inflammation and negatively regulates M2 macrophage polarization, and the overexpression of DNMT3b inhibits arginase-1 (Arg-1) expression under IL-4 stimulation [6]. Meanwhile, a high level of DNMT3b promotes M1 macrophage polarization [7]. Macrophage polarization plays a key role in the inflammatory process. M1 macrophages (classically activated macrophages), which mainly enhance inflammation, are defined as proinflammatory macrophages, whereas those that decrease inflammation are called M2 macrophages (alternatively activated macrophages) [8,9]. Overall, the high expression of DNMT3b is positively correlated with inflammation, suggesting that DNMT3b-regulated macrophage phenotypes participate in the regulation of inflammation [10].

A previous study indicated that the suppression of the signal transducer and activator of the transcription 3 (STAT3) pathway can effectively inhibit inflammatory reactions [11]. It has been reported that the expression of DNMT3b is regulated by the interleukin 6/phosphorylated STAT-3 (IL-6/p-STAT-3) pathway, and activating this pathway can increase the level of DNMT3b [12]. The knockout of DNMT3b inhibits the expression of the inflammatory factor TNF-α [8].

The nuclear factor erythroid 2-related factor 2 (Nrf2) pathway is also considered to inhibit the progression of inflammation when it is activated [13], and antioxidant gene expression belonging to the Nrf2 pathway is considered to reduce classically activated macrophages [14,15]; therefore, the Nrf2 pathway also plays a very important role in inflammation-related diseases.

Recently, many studies have shown that natural ingredients derived from different plants have anti-inflammatory effects [16,17]. *Dipsacus asper* Wall. ex Henry (DAW) is a traditional Chinese medicine with anti-rheumatic arthritis effects [18]. Akebia saponin D (ASD) is a natural monomeric compound (Appendix A) derived from DAW and is also called asperosaponin VI [19]. Scientific research has shown that ASD can treat inflammatory diseases [20]. However, the mechanism of action of ASD targets is still unclear. In this study, we explored the anti-inflammatory mechanism of ASD from the regulation of DNMT3b and related pathways using a lipopolysaccharide (LPS)-stimulated RAW264.7 cell model. We found that (1) ASD was able to inhibit inflammatory reactions in a macrophage inflammation model, (2) ASD inhibited DNMT3b expression to regulate macrophage polarization and exert its anti-inflammatory action, and (3) ASD inhibited the inflammatory reaction by inhibiting the IL-6-STAT3-DNMT3b axis while simultaneously activating the Nrf2/HO-1 pathway.

## 2. Materials and Methods

### 2.1. Materials

ASD (HPLC purity > 98%) was purchased from Chengdu Alfa Biotechnology Co., Ltd. (Chengdu, China). LPS was purchased from Beijing Solabao Technology Co. (Beijing, China). The antibodies against iNOS, p-STAT3, HO-1, and Nrf2 were obtained from Wuhan Sanying Biotechnology Co., Ltd. or Affinity Biosciences. DNMT3b was purchased from Abcam. ELISA kits for PGE_2_ were obtained from Cayman Chemical; ELISA kits detecting IL-6 and TNF-α were obtained from MultiSciences (Lianke) Biotech Co., Ltd.; siRNA for Nrf2 (sc-37049) and control siRNA (sc-37007) were obtained from Santa Cruz Biotechnology; and the antibody against β-actin was obtained from Wuhan Sanying Biotechnology Co., Ltd. siRNA for DNMT3b was obtained from RiboBio (Guangzhou, China). The secondary antibodies were purchased from Abixin Biotechnology Co., Ltd. (Shanghai, China) or Wuhan Bode Bioengineering Co., Ltd. (Wuhan, China). The protein molecular markers were obtained from GenScript ProBio or BIO-RAD; SYBR Green master mix was obtained from Roche, Ltd.; OPTI-MEM medium was obtained from Invitrogen; F/480 and CD11c were obtained from BioLegend, San Diego, CA, USA; the XtremeGENE™ HP DNA transfection reagent was obtained from Roche; the Bradford assay reagent and the RNeasy Mini Kit were obtained from Beijing Solabao Technology Co.; and the reverse transcription cDNA kit was obtained from Roche, Ltd.

### 2.2. Cell Culture

RAW264.7 cells were obtained from Wuhan Punuosai Life Technology Co., Ltd. The cells were cultured in full Dulbecco’s modified Eagle’s medium (DMEM); this full DMEM contained 100 mg/mL streptomycin, 100 U/mL penicillin G, and 10% FBS. The cells were incubated in a cell incubator at 37 °C with 5% CO_2_.

### 2.3. MTT Assay

The MTT method was used to detect cell viability. In brief, RAW264.7 macrophages were seeded in ninety-six-well plates (1.4 × 10^4^ cells per well) overnight and then incubated with ASD (1.25, 2.5, 5, 10, 20, 40, and 80 μM) for 18 h (without or with 100 ng/mL LPS). Ten microliters of MTT solution (5 g/L) were added to each well, and the cells were cultured at 37 °C for another 4 h. Then, 200 μL of 10% SDS-HCl solution were added to each well. Finally, the OD value was obtained at a 560/650 wavelength.

### 2.4. Measurement of the PGE_2_, NO, IL-6 and TNF-α Levels

RAW264.7 cells were seeded (80,000 cells per well) and incubated in a twenty-four-well plate overnight. The cells were incubated with ASD for one hour and then induced by 100 ng/mL LPS for another eighteen hours. The production of PGE_2_, NO, IL-6, and TNF-α was detected by following the instructions of each ELISA Kit.

### 2.5. Real-Time PCR Analysis

RAW264.7 cells were seeded (80,000 cells per well) and incubated overnight. The cells were pretreated with ASD (5, 10, and 20 μM) for one hour and then incubated with 100 ng/mL LPS for 6 h to detect Arg-1 and SOCS2 gene expression or incubated with 100 ng/mL LPS for 18 h to detect other target gene levels. The total RNA was extracted by using a kit obtained from Beijing Solabao Technology Co. The total RNA purity and concentration were determined by measuring the absorbance at 260 and 280 nm. One microgram of total RNA was reverse synthesized into cDNA (Roche, Mannheim, Germany). The PCR system included 1 μL cDNA, 10 μL SYBR Green PCR Master Mix (Roche, Mannheim, Germany), 2 μL primers (Huada gene, Guangzhou, China), and 7 μL PCR-grade water (the final total reaction volume was 20 μL). After mixing, the reaction mixture was placed in a 7500 Real-Time PCR System (Applied Biosystems, Foster City, CA, USA). The reactions were performed with a denaturation step at 95 °C for 10 min, followed by a cycle of 40 times of 95 °C for 15 s and 60 °C for 1 min.

The internal reference gene was β-actin, and the expression of each target gene was normalized to β-actin. The expression of each target gene was calculated by using the 2^−ΔΔCt^ method. The relative target gene expression was detected by using ViiATM 7 real-time PCR. The primer sequences are listed in Table 1.

### 2.6. Western Blot Analysis

Cells were seeded in a twenty-four-well plate (80,000 cells per well) and incubated overnight. The cells were treated with ASD (5, 10, and 20 μM) for one hour and then incubated with 100 ng/mL LPS for 18 h. The cells were collected by using the RIPA lysis buffer, and the Bradford assay reagent was used to detect the protein concentration. The protein samples were separated by SDS–PAGE using the same amount and then transferred to a nitrocellulose membrane. A 5% BSA solution or 5% skimmed milk solution was used to block the membrane, which was then incubated with the target primary antibody (1:1000) overnight at 4 °C. Then, secondary antibodies (1:10,000) were used to treat the membranes for 1 h at room temperature. An Odyssey CLx Imager (Li-COR, Lincoln, NE, USA) or electrophoresis gel, SDS–PAGE, and Western blotting films were used to scan the band of antigen-antibody complexes. ImageJ software was used for the analysis of the results.

### 2.7. Flow Cytometric Analysis

The macrophage M1 subtype was analyzed by using flow cytometry. The cells were seeded in a six-well cell plate (3 × 10^5^ cells per well) for 18 h and then pretreated with ASD (20 μM) for 1 h after stimulation with LPS (100 ng per mL) for 6 h. The above cells were stained with a flow cytometry antibody for the flow cytometric analysis by using a BD FACSAria III Flow Cytometer (BMSE-001), and the final results were analyzed by using FlowJo software.

### 2.8. Transfection Assays

DNMT3b siRNA, Nrf2 siRNA, and control siRNA were cultured with RAW264.7 cells by using XtremeGENE™ HP DNA transfection reagent according to the kits’ instructions. Next, the transfected cells were preconditioned with ASD for 1 h and cultured with or without LPS (100 ng per mL) for another 18 h. The expression of DNMT3b, Nrf2, and HO-1 was detected by RT–PCR or a Western blot analysis.

### 2.9. Statistical Analysis

The data are expressed as the average ± SD (N = 3). The statistical analysis was performed by using a one-way ANOVA, followed by a post hoc analysis for multiple comparisons; in GraphPad Prism 7, *p* < 0.05 was considered statistically significant.

## 3. Results

### 3.1. ASD Inhibited the Expression Levels of NO and PGE_2_ in LPS-Treated Macrophages

The MTT results show that a treatment concentration of ASD from 1.25 to 20 μM is a safe dosage for RAW264.7 cells (Figure 1B,C), regardless of whether the cells are incubated with or without LPS. Therefore, 5 μM to 20 μM ASD were used in the following experiments. In the inflammatory response, NO was induced (under the induction of LPS) [21]. When the cells were incubated with LPS for 18 h, NO was significantly increased (Figure 1C). ASD obviously reduced the level of NO in the LPS-treated macrophages (Figure 1C). The production level of PGE_2_ was significantly increased after the LPS stimulation (Figure 1D). ASD also obviously reduced the level of PGE_2_ in the LPS-treated macrophages (Figure 1D).

### 3.2. ASD Restrained the Levels of iNOS, DNMT3b, TNF-α, and IL-6

During inflammation, activated macrophages release abundant TNF-α and IL-6 to aggravate the inflammatory response [22]. In addition, a study found that the expression of DNMT3b is increased under inflammatory conditions [10] and inhibiting the expression of DNMT3b can promote anti-inflammatory effects [23]. Under inflammatory conditions, NO is synthesized by iNOS [21]. Figure 2A,B demonstrates that the iNOS and DNMT3b protein levels were elevated in the LPS-induced cells, while the increased iNOS and DNMT3b protein levels were reduced after the pretreatment with ASD. As shown in Figure 2C,D, the iNOS and DNMT3b mRNA levels were increased after the LPS stimulation. Meanwhile, as shown in Figure 2E,F, LPS increased the production of TNF-α and IL-6, while ASD significantly decreased the levels of TNF-α and IL-6 (Figure 2E,F).

### 3.3. The Effect of ASD on the Expression Levels of M1, TNF-α, IL-6, Arg-1, and SOCS2

The regulation of inflammation is significantly influenced by macrophage polarization; M1 macrophages have a proinflammatory effect, while M2 macrophages can inhibit inflammatory reactions [8]. As shown in Figure 3A,B, the M1 phenotype was increased in the LPS-treated cells, and this increase was suppressed by the ASD pretreatment. In addition, the ASD pretreatment reduced the elevated levels of TNF-α and IL-6 (Figure 3C,D). As shown in Figure 3E,F, the ASD pretreatment increased the levels of the Arg-1 gene (Figure 3E) and SOCS2 gene (Figure 3F) in the IL-4-stimulated RAW264.7 cells in a dose-dependent manner.

### 3.4. The Effect of ASD on the Levels of p-STAT3 and HO-1

The inhibition of the STAT3 pathway is thought to regulate the inflammatory reaction [24]. As shown in Figure 4A, the level of p-STAT3 was increased under the LPS stimulation. Meanwhile, the increased level of the p-STAT3 protein was inhibited by the ASD pretreatment (Figure 4A). The results also show that the ASD pretreatment dramatically increased the HO-1 protein levels (Figure 4B), while LPS only slightly increased the HO-1 protein levels (Figure 4B).

### 3.5. DNMT3b siRNA Significantly Promoted the Anti-Inflammatory Effect of ASD

To further prove the key role of DNMT3b in the ASD effect, the DNMT3b siRNA transfection method was used to knockdown the DNMT3b gene. The results show that the DNMT3b gene and protein were knocked down by using the DNMT3b siRNA transfection approach (Figure 5A,B). The increased DNMT3b mRNA and protein levels were downregulated by DNMT3b siRNA (Figure 5A,B). The suppressive effect of ASD on NO and TNF-α was also partially blocked by the DNMT3b siRNA treatment in LPS-induced cells (Figure 5C,D).

### 3.6. Nrf2 siRNA Significantly Abolished the Effect of ASD on NO and HO-1

Nrf2-mediated antioxidant gene expression promotes anti-inflammatory effects [14]. HO-1 antioxidant protein, which is downstream of the Nrf2 pathway, has an anti-inflammatory effect [25,26]. To prove the key effect of Nrf2 on the effect of ASD, Nrf2 siRNA was used to knockdown the Nrf2 gene. The results show that the expression of Nrf2 at both the gene and protein levels was significantly downregulated by the Nrf2 siRNA pretreatment (Figure 6A,B). The downregulatory effect of ASD on NO was also blocked by using Nrf2 siRNA (Figure 6C). The increase in HO-1 at the mRNA level by the ASD pretreatment was blocked by Nrf2 siRNA (Figure 6D).

## 4. Discussion

Activated macrophages release PGE_2_, TNF-α, NO, IL-6, and regulatory enzymes (iNOS) under inflammatory conditions [25,26,27,28,29,30,31]. In this research, the levels of NO, PGE_2_, TNF-α, IL-6, and iNOS were increased in LPS-activated RAW264.7 macrophages, but these increased inflammatory markers were significantly inhibited by the ASD pretreatment (Figure 1 and Figure 2), suggesting that ASD has the ability to inhibit inflammatory reactions in LPS-induced RAW264.7 macrophages.

Chronic inflammation is accompanied by high levels of DNA methylation, and previous studies suggested that the activation of inflammation causes abnormal DNA methylation by activating the expression of the DNA methylation enzyme DNMT3b [32]. It has been reported that DNMT3b expression is regulated by the IL-6/p-STAT-3 pathway, and the activation of the IL-6/p-STAT-3 pathway increases the level of DNMT3b [12]. Furthermore, the knockout of DNMT3b has the ability to inhibit the expression of TNF-α [6]. Inhibiting the expression of DNMT3b has been shown to inhibit the inflammatory reaction [23]. The results show that ASD inhibited the high levels of iNOS and DNMT3b at both the protein and gene levels in LPS-induced RAW264.7 macrophages (Figure 2A,D). In addition, ASD inhibited the high expression of TNF-α and IL-6 at the protein level in LPS-treated RAW264.7 macrophages (Figure 2E,F).

Macrophage polarization is closely related to inflammatory reactions. TNF-α and IL-6 are markers of M1 macrophages, and Arg1 is a marker of M2 macrophages. Another study suggested that increased SOCS2 may promote M2 polarization [33]. This study found that ASD significantly inhibited the expression of M1 macrophage markers (iNOS, TNF-α, and IL-6); therefore, this study suggested that ASD significantly prevented M1 macrophages from playing an anti-inflammatory role (Figure 3A,D). However, this study found that ASD can increase the expression of Arg-1 at the gene level. As Arg-1 is a marker of M2 macrophages, M2 macrophages are considered to have an anti-inflammatory effect; therefore, this finding indicates that ASD significantly promoted M2 macrophages, which may play an anti-inflammatory role (Figure 3E). The mechanistic study found that ASD increased the expression of SOCS2, which contributed to M2 polarization (Figure 3F).

A previous report indicated that the inhibition of iNOS, TNF-α, and IL-6 inhibited the activation of the STAT3 pathway [34]. To explore the mechanism of ASD, the effects of ASD on the STAT3 pathway were investigated in LPS-induced RAW264.7 macrophages. The results suggest that ASD inhibited the activation of the p-STAT3 protein, which is a key protein in the activation of the STAT3 pathway (Figure 4A). This result indicates that ASD may act by inhibiting the STAT3 pathway to contribute to its anti-inflammatory effects. IL-6 is a potent activator of STAT3, and STAT3 is known to upregulate DNMT3b expression [35]. The results show that ASD inhibited the STAT3 pathway (Figure 3A) and inhibited the expression of IL-6 (Figure 2F) and DNMT3b (Figure 2B) from playing an anti-inflammatory role. Overall, ASD may act by inhibiting the IL-6/STAT3/DNMT3b axis to achieve its anti-inflammatory effects.

The antioxidant gene HO-1, which is downstream of the Nrf2 pathway, can block inflammatory factors. Under inflammatory conditions, increased HO-1 contributes to cell protection [36,37]. A high level of HO-1 can inhibit the level of NO in LPS-induced cells [38]. A high expression of HO-1 promotes M2 polarization [39], and a high level of HO-1 inhibits M1 polarization [40]. The current results also show that ASD dramatically increased the HO-1 protein (Figure 4B). This study found that ASD promoted the level of HO-1, which may be one of the mechanisms by which ASD inhibits M1 activation and promotes M2 polarization, resulting in its anti-inflammatory effect.

To further prove that DNMT3b plays a key role in the effect of ASD, DNMT3b siRNA was used to decrease the gene and protein expression of DNMT3b (Figure 5A,B). The results indicate that DNMT3b siRNA can increase the inhibitory effect of ASD on NO and TNF-α production in LPS-induced RAW264.7 macrophages (Figure 5C,D). These results prove that the effect of ASD on the inflammatory reaction was partially mediated by inhibiting DNMT3b. The knockout of the DNMT3b gene (Figure 5C,D) inhibited inflammatory mediators (NO and TNF-α). The above results prove that ASD played an anti-inflammatory role by partially inhibiting DNMT3b.

Studies have shown that DNMT3b is closely related to macrophage polarization, and the knockout of DNMT3b can promote RAW264.7 cell polarization to M2 polarization. The knockout of the DNMT3b gene can also inhibit the expression of inflammatory factors [6]. Furthermore, it was found that the knockout of the DNMT3b gene further promoted the high level of Arg-1 induced by IL-4 and the ASD treatment (Figure 5E) in this research, indicating that the inhibitory effect of ASD on DNMT3b can promote the M2 polarization of macrophages. These results confirm the important role of DNMT3b in the anti-inflammatory effect of ASD.

Once the Nrf2 pathway is activated, it can inhibit proinflammatory cytokines. It is worth studying whether ASD activates this pathway to play a therapeutic role. In this paper, Nrf2 siRNA was used to decrease the levels of the Nrf2 gene and protein expression. The results indicate that Nrf2 siRNA decreased the Nrf2 gene and protein levels (Figure 6A,B). This research shows that Nrf2 siRNA abolished the effect on HO-1 and NO induced by the ASD treatment in LPS-stimulated macrophages (Figure 6C,D). This study proves that ASD can activate the Nrf2/HO-1 pathway, which helps exert some of the anti-inflammatory effects of ASD.

In summary, ASD inhibited the activation of the IL-6/STAT3 pathway to reduce the expression of DNMT3b, thereby achieving its anti-inflammatory effect. In addition, ASD activated the Nrf2 pathway to upregulate the HO-1 levels, which, in turn, reduced inflammation. In summary, inhibiting the IL-6-STAT3-DNMT3b axis contributes to the anti-inflammatory effects of ASD, and regulating macrophage polarization and activating the Nrf2 pathway in LPS-stimulated RAW264.7 macrophages contributes to the anti-inflammatory effects induced by the ASD treatment (Figure 7). This study provides a new potential treatment mechanism of ASD for the treatment of inflammation and abnormal DNA methylation-related diseases.

## Figures and Tables

**Figure 1 molecules-27-06236-f001:**
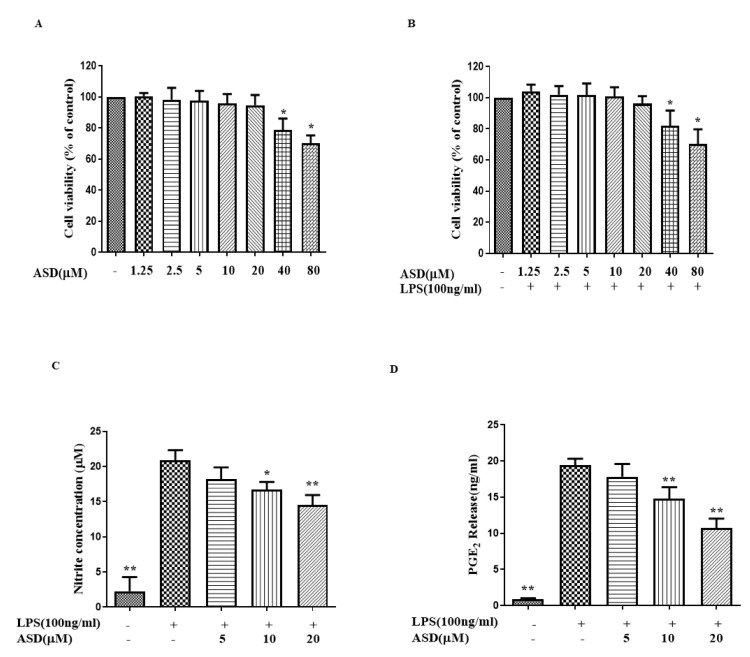
The pharmacological action of ASD on the levels of NO and PGE_2_. (**A**) Effect of ASD toxicity on the cell viability (without LPS). (**B**) Effect of ASD toxicity on the cell viability (with LPS). Effects of ASD on the secretion of NO (**C**) and PGE_2_ (**D**). * *p* < 0.05, vs. LPS-stimulated cells (**A**,**B**). The data are shown as the average value ± SD, * *p* < 0.05, vs. LPS-treated cells, ** *p* < 0.01, vs. LPS-treated cells (**C**,**D**).

**Figure 2 molecules-27-06236-f002:**
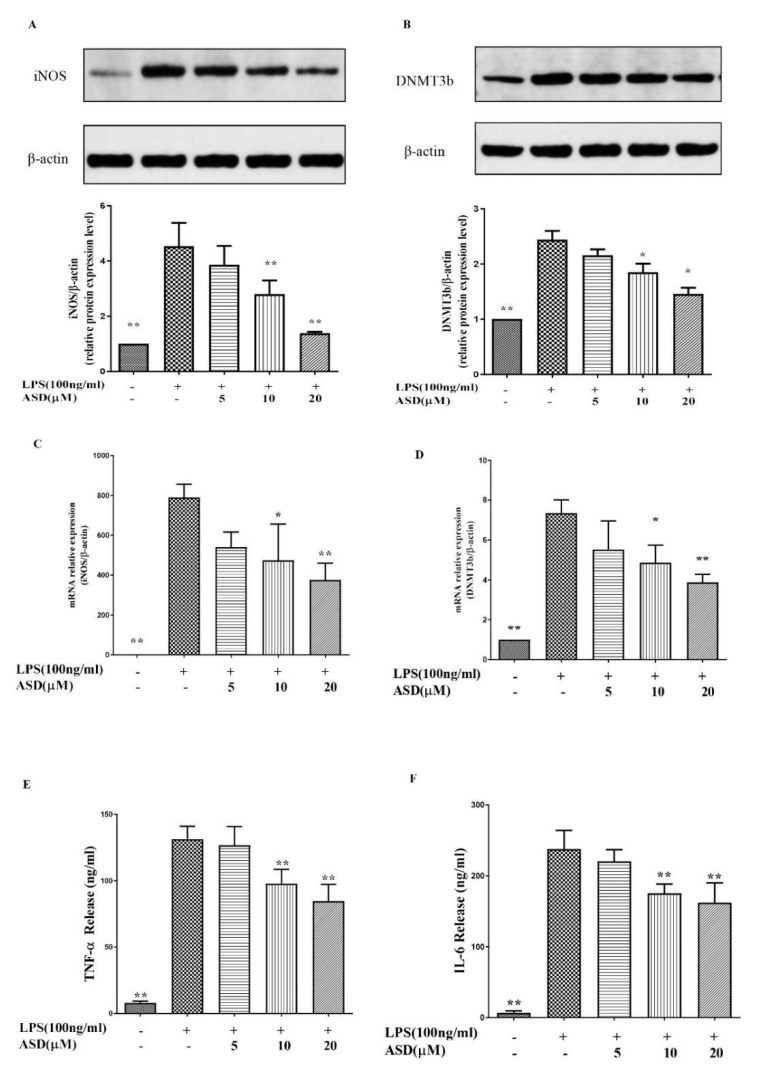
Pharmacological effect of ASD on the levels of iNOS, DNMT3b, TNF-α, and IL-6. The protein expression levels of iNOS (**A**) and DNMT3b (**B**). The gene levels of iNOS (**C**) and DNMT3b (**D**). The protein expression levels of TNF-α (**E**) and IL-6 (**F**). The data are shown as the average value ± SD, * *p* < 0.05, vs. LPS-treated cells, ** *p* < 0.01, vs. LPS-treated cells.

**Figure 3 molecules-27-06236-f003:**
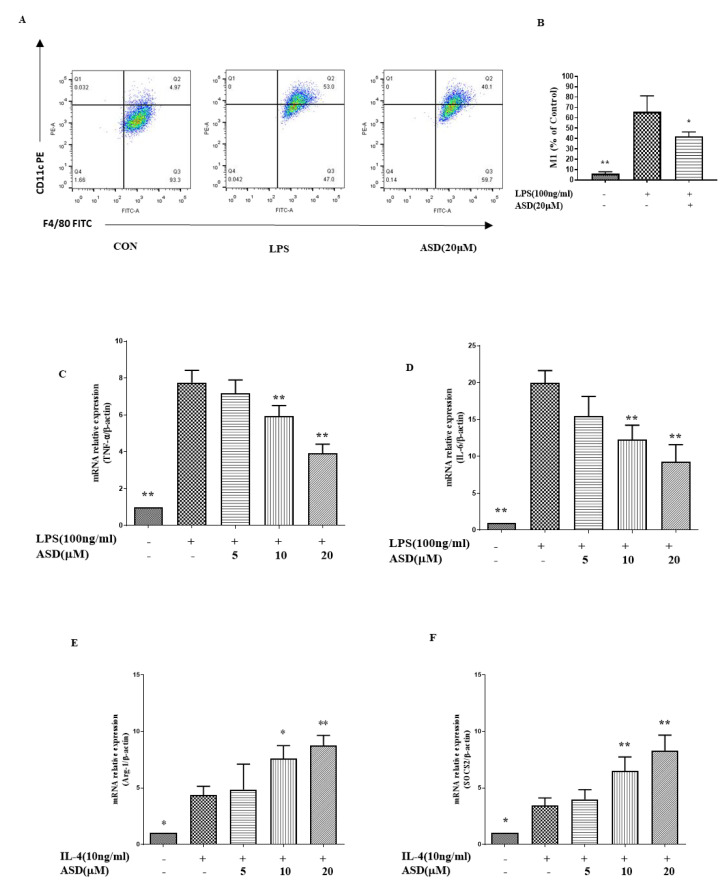
Pharmacodynamic effect on the expression of M1, TNF-α, IL-6, Arg-1, and SOCS2. The M1 markers (**A**,**B**) were detected following the kits’ methods. Gene expression levels of TNF-α (**C**), IL-6 (**D**), Arg-1 (**E**), and SOCS2 (**F**). The data are shown as the average value ± SD, * *p* < 0.05, vs. LPS-treated cells, ** *p* < 0.01, vs. LPS-treated cells (**B**,**D**). * *p* < 0.05, vs. IL-4-treated cells, ** *p* < 0.01, vs. IL-4-treated cells (**E**,**F**).

**Figure 4 molecules-27-06236-f004:**
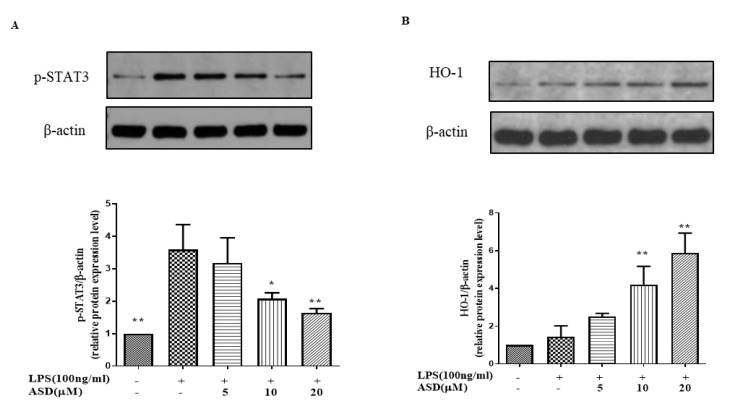
The effect of ASD on the protein levels of p-STAT3 and HO-1. Expression of p-STAT3 (**A**) and HO-1 (**B**). The data are shown as the average value ± SD, * *p* < 0.05, vs. LPS-treated cells, ** *p* < 0.01, vs. LPS-treated cells.

**Figure 5 molecules-27-06236-f005:**
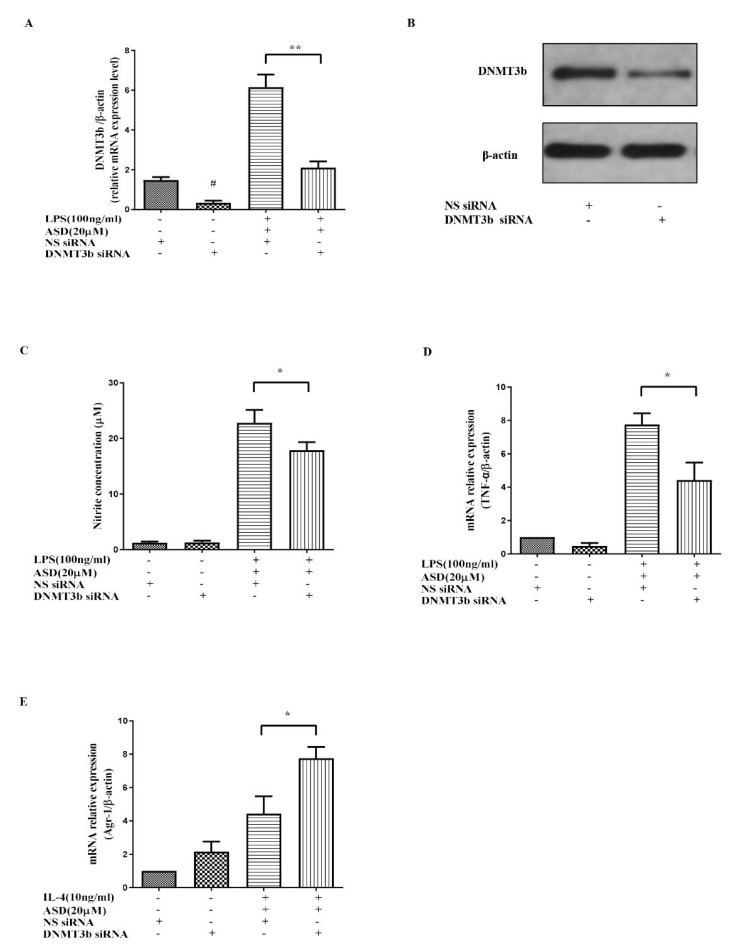
The effect of DNMT3b siRNA on the effect of ASD. The effect of DNMT3b siRNA on the expression of DNMT3b (**A**,**B**), NO (**C**), TNF-α (**D**), and Arg-1 (**E**). The data are shown as the average value ± SD. # *p* < 0.05 vs. NS siRNA-treated cells *, *p* < 0.05, vs. LPS and NS siRNA-treated cells, ** *p* < 0.01, vs. LPS and NS siRNA-treated cells (**A**–**E**).

**Figure 6 molecules-27-06236-f006:**
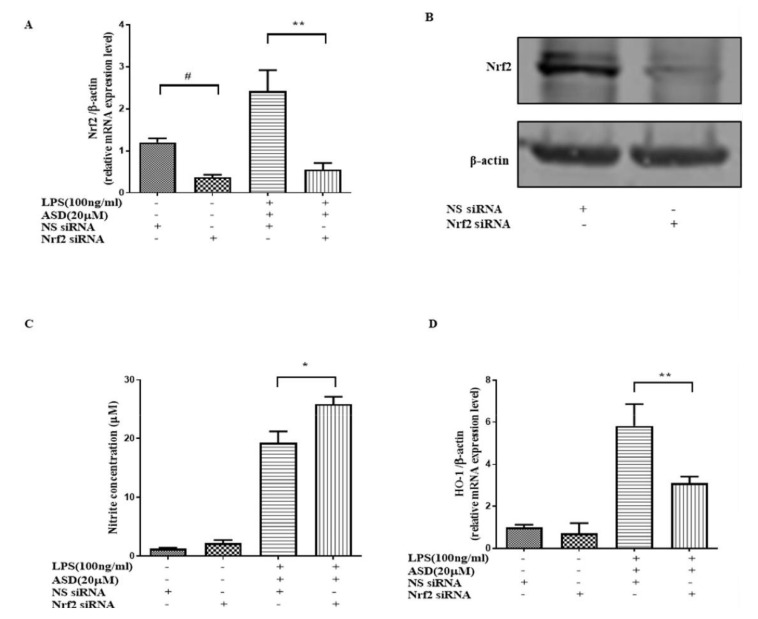
The effect of Nrf2 siRNA on the effect of ASD. The effect of Nrf2 siRNA on the expression levels of Nrf2 (**A**,**B**), NO (**C**), HO-1 (**D**), and Arg-1 (**E**). The data are shown as the average value ± SD. # *p* < 0.05 vs. NS siRNA-treated cells (**A**), * *p* < 0.05, vs. LPS and NS siRNA-induced cells (**A**–**D**), ** *p* < 0.01, vs. LPS and NS siRNA-induced cells (**A**–**D**).

**Figure 7 molecules-27-06236-f007:**
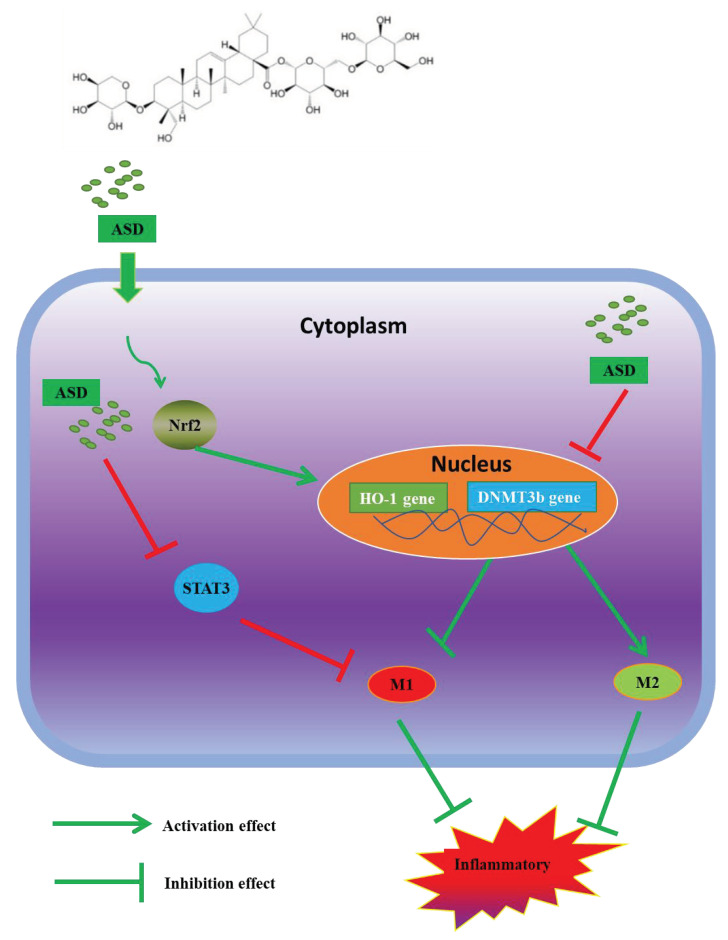
Diagram of the proposed molecular mechanism by which ASD activates RAW264.7 macrophages (under LPS stimulation). ASD inhibits the inflammatory reaction by inhibiting the IL-6-STAT3-DNMT3b axis and simultaneously activates the Nrf2/HO-1 pathway.

**Table 1 molecules-27-06236-t001:** The primers sequenses for RT-PCR.

Target Gene	Primer Sequences
β-actin_F	5′-CGGTTCCGATGCCCTGAGGCTCTT-3′
β-actin_R	5′-CGTCACACTTCATGATGGAATTGA-3′
iNOS_F	5′-CAGCACAGGAAATGTTTCAGC-3′
iNOS_R	5′-TAGCCAGCGTACCGGATGA-3′
DNMT3b_F	5′-TTCAGTGACCAGTCCTCAGACACGAA-3′
DNMT3b_R	5′-TCAGAAGGCTGGAGACCTCCCTCTT-3′
TNF-α_F	5′-TATGGCTCAGGGTCCAACTC-3′
TNF-α_R	5′-CTCCCTTTGCAGAACTCAGG-3′
IL-6_F	5′-GGTGACAACCACGGCCTTCCC-3′
IL-6_R	5′-AAGCCTCCGACTTGTGAAGTGGT-3′
Nrf2_F	5′-AGCAGGACATGGAGCAAGTT-3′
Nrf2_R	5′-TTCTTTTTCCAGCGAGGAGA-3′
HO-1_F	5′-CCCACCAAGTTCAAACAGCTC-3′
HO-1_R	5′-AGGAAGGCGGTCTTAGCCTC-3′
Arg-1_F	5′-AGCTCTGGGAATCTGCATGG-3′
Arg-1_R	5′-ATGTACACGATGTCTTTGGCAGATA-3′
SOCS2_F	5′-CTGCGCGAGCTCAGTCAAAC-3′
SOCS2_R	5′-CAAGAAAGTTCCTTCTGGAGCCTCT-3′

## Data Availability

The data presented in this study are available upon request from the corresponding author.

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
