# Peer review of "Akebia Saponin D Inhibits the Inflammatory Reaction by Inhibiting the IL-6-STAT3-DNMT3b Axis and Activating the Nrf2 Pathway"

_molecules, 2022, doi:10.3390/molecules27196236_

Round 1

Reviewer 2 Report

The author should provide the 1H-NMR of Akebia Saponin D for characterization profiling in the Supplementary material.

Author Response

Thank you for your suggestions. The 1H-NMR of Akebia Saponin D has been added in the supplementary material.

Reviewer 3 Report

The current work is objective and clear and well-linked. However, the discussion needs to be revised, some sentences have repeated words and redundancy or do not add information, as can be seen in the lines:

2019-2020 - remove the phrase "Overactivated macrophages produced inflammatory mediators, then exacerbated inflammation [22]." it adds no information;

228 - remove "On one hand..."

230 - remove "belonging to M1 macrophage..."

293, 294 and 295 - to improve the way of presenting the results in the discussion, the word the expression "the results" appears a lot.

I think it's important to put the chemical structure of the ASD.

I believe that ASD has an inhibitory effect on macrophages stimulated by LPS, however, in most results the inhibition is less than 45%. To affirm an anti-inflammatory effect, it is necessary to evaluate the action of ASD on the IL-6-STAT3-DNMT3b axis in vivo assays.

Reviewer 4 Report

The authors work to present the pathway with which the compound Akebia Saponin D (ASD) interacts. While a potentially interesting piece, it requires some further consideration.

The materials and methods section is lacking and the results cannot be reproduced from the information presented. For RT-PCR, MIQE guidelines are not followed. There is information missing from the blotting experiments (e.g., what dilution of antibody was used), and the section on flow cytometry fails to name a machine or any type of technique. There isn’t even mention of how the authors did densitometry analysis. At best, this is an incomplete section that requires significant additions.

The authors fail to include a significant control in all of their experiments in the second half of figure 3 and all of figure 4, which constitute the majority of the paper. Without an ASD control without the addition of LPS (all instances of ASD addition have LPS as well), we cannot conclude that the influence of ASD is truly a positive one. For example, the highly increased levels of TNF-alpha or IL-6 in the presence of LPS are expected, but they stay really high with the addition of ASD. What is the influence of ASD alone on TNF-alpha or IL-6? Without any sort of comparison, the authors cannot claim much about the actual influence of ASD on the process and certainly cannot state that a mechanism has been determined.

While it may be true that the addition of ASD resulted in “significant” inhibition of the LPS response, the data indicate that the level of attenuating LPS-mediated spikes in inflammation responses were low. There was almost never a return to baseline in any of the experiments so, while some interaction of ASD may have happened, the result was minor. In the case of TNF-alpha or IL-6, the addition of 10 uM or 20 uM of ASD does not result in an additional decrease in LPS-mediated increases at the higher concentration. If ASD is doing something in a stoichiometric fashion (as we expect given the presentation of the graphs with increasing ASD concentrations), we would expect to see a continued trend in decreasing the LPS-mediated increases in TNF-alpha or IL-6. While there does appear to be some result, it is not the slam dunk that the title/abstract indicates and, though some results were generated, this does not conclusively demonstrate the pathway utilized. These are all correlational responses because ASD could influence some other component that interacts with the named pathways. To demonstrate pathways, the authors would use other known compounds that also influence these pathways and demonstrate that the 1) the results are the same as those with compounds known to inhibit these pathways or 2) that the known compounds inhibit the function of ASD, which can again demonstrate causality in a response pathway. Additionally, cell lines (or organismal lines) mutant for parts of these pathways could be interrogated to demonstrate that the compound interacts with the pathway. As no additional confirmations were used, this work does not live up to the claim in the abstract: “This study aimed to systematically explore the anti-inflammatory effect of ASD and the underlying mechanisms from the perspective of DNA methylation.”

Finally, the writing requires extensive editing and, in some cases, the word choice obscures the science.

Author Response

Reviewer #4 comments and our response:

Reviewer #4 comments

The authors work to present the pathway with which the compound Akebia Saponin D (ASD) interacts. While a potentially interesting piece, it requires some further consideration.

The materials and methods section is lacking and the results cannot be reproduced from the information presented. For RT-PCR, MIQE guidelines are not followed. There is information missing from the blotting experiments (e.g., what dilution of antibody was used), and the section on flow cytometry fails to name a machine or any type of technique. There isn’t even mention of how the authors did densitometry analysis. At best, this is an incomplete section that requires significant additions.

Our response:

Thank you for your suggestions.

We have modified the manuscript based on your valuable modifications point by point. Our responses are described below one by one.

(1) The detailed experimental information of RT-PCR experiment has been supplemented according to the comments (line 122-138).

(2) Diluted concentration of supplemental antibody has been supplemented according to the comments (line 148-149).

(3) The section on flow cytometry has been supplemented according to the comments (line157-159).

Reviewer #4 comments

The authors fail to include a significant control in all of their experiments in the second half of figure 3 and all of figure 4, which constitute the majority of the paper. Without an ASD control without the addition of LPS (all instances of ASD addition have LPS as well), we cannot conclude that the influence of ASD is truly a positive one. For example, the highly increased levels of TNF-alpha or IL-6 in the presence of LPS are expected, but they stay really high with the addition of ASD. What is the influence of ASD alone on TNF-alpha or IL-6? Without any sort of comparison, the authors cannot claim much about the actual influence of ASD on the process and certainly cannot state that a mechanism has been determined.

Our response:

Thank you for your suggestions. Other previous studies[1-3]  have  reported that ASD has a definite anti-inflammatory effect and can significantly inhibit the production of inflammatory factors. What is the influence of ASD alone on TNF-alpha or IL-6? we did not make a separate group with only ASD treated cells. As in the absence of LPS stimulation, cells do not overproduce inflammatory cytokines(including TNF-alpha or IL-6),the addition of ASD to normal cells can not explain whether drugs have an inhibitory effect on LPS activated inflammatory RAW264.7 cells. At present, this study mainly focuses on whether ASD have inhibitory effects on LPS activated inflammatory RAW264.7 cells and the possible mechanism. Therefore, it is mainly to compare the effects of ASD on LPS induced cells. In future experiments, we will only add ASD control without the addition of LPS according to your suggestion to see whether the ASD has any effect on normal cells.

Reviewer #4 comments

While it may be true that the addition of ASD resulted in “significant” inhibition of the LPS response, the data indicate that the level of attenuating LPS-mediated spikes in inflammation responses were low.

Our response:

Thank you for your suggestions. The ASD described in this study inhibited inflammation related markers is an experimental conclusion based on statistical analysis of experimental results (P < 0.01 or P < 0.05 vs LPS induced cells).

Reviewer #4 comments

There was almost never a return to baseline in any of the experiments so, while some interaction of ASD may have happened, the result was minor. In the case of TNF-alpha or IL-6, the addition of 10 μM or 20 μM of ASD does not result in an additional decrease in LPS-mediated increases at the higher concentration.

Our response: Thank you for your suggestions. Our previous anti-inflammatory experimental study also used 100ng / ml LPS to establish a cellular inflammation model [1]. It can also be reported in other literatures [2, 3] and other drugs have inhibitory effects on inflammatory factors (the inhibitory effect of other drugs on TNF alpha or IL-6 did not return to baseline). However, the results of those papers also showed that the drugs have anti-inflammatory effects.

The literature reported [4] that ASD at 25μM significantly inhibited the NO production induced by LPS. This result indicates that ASD also has a clear anti-inflammatory effect under the stimulation of high concentration of LPS.

In the future, we will conduct experiments with high concentration of LPS according to your suggestions to study the anti-inflammatory effect and mechanism of ASD.

Reviewer #4 comments

If ASD is doing something in a stoichiometric fashion (as we expect given the presentation of the graphs with increasing ASD concentrations), we would expect to see a continued trend in decreasing the LPS-mediated increases in TNF-alpha or IL-6. While there does appear to be some result, it is not the slam dunk that the title/abstract indicates and, though some results were generated, this does not conclusively demonstrate the pathway utilized.

Our response:

Thank you for your suggestions. The reason why the anti-inflammatory experiment was not continued to increase the concentration of ASD was that when the concentration of ASD above 40μM has a certain inhibitory effect on the cell viability, so the high concentration of drugs was not selected for the experiment, and the low concentration without obvious inhibition on the cell viability was selected for the experiment in this study.

Reviewer #4 comments

These are all correlational responses because ASD could influence some other component that interacts with the named pathways. To demonstrate pathways, the authors would use other known compounds that also influence these pathways and demonstrate that the 1) the results are the same as those with compounds known to inhibit these pathways or 2) that the known compounds inhibit the function of ASD, which can again demonstrate causality in a response pathway. Additionally, cell lines (or organismal lines) mutant for parts of these pathways could be interrogated to demonstrate that the compound interacts with the pathway.

Our response: Thank you for your suggestions. Previous study indicated that the suppression of signal transducer and activator of transcription 3 (STAT3) pathway can effectively inhibit inflammatory reaction[5] . It has reported that the expression of DNMT3b is regulated by interleukin 6/phosphorylated STAT-3 (IL-6/p-STAT-3) pathway, and activating this pathway has the ability to increase the level of DNMT3b [6]. Knockout of DNMT3b inhibits the expression of inflammatory factor TNF-α [7].

In this study, we also silenced DNMT3b gene with siRNA, and the results also showed that silencing DNMT3b gene increased the effect of ASD on TNF- α. The results showed that the anti-inflammatory effect of ASD was partly caused by the downregulation of DNMT3b by ASD. In addition, LPS induced the high expression of IL-6 and p-STAT3, and ASD downregulated the high expression of IL-6 and p-STAT3 induced by LPS. Our results preliminarily indicate that ASD inhibited IL-6-STAT3-DNMT3b axis to play its anti-inflammatory effect. Not all pathway studies require the use of pathway inhibitors,the pathway related literature[[8]]  also does not use pathway inhibitors. In this study, we used Nrf2 siRNA to silence the Nrf2 protein. The results also showed that silencing Nrf2 gene weaken the anti-inflammatory effect of ASD. Therefore, it can partially prove that ASD exerts its anti-inflammatory effect through acting on Nrf2 pathway.

Indeed, it would be better if the experimental study could be conducted with pathway inhibitors as you suggested. We will follow your suggestion to use pathway inhibitors in future studies.

Reviewer #4 comments

As no additional confirmations were used, this work does not live up to the claim in the abstract: “This study aimed to systematically explore the anti-inflammatory effect of ASD and the underlying mechanisms from the perspective of DNA methylation.”

 Our response: Thank you for your modification suggestion. The claim in the abstract: “This study aimed to systematically explore the anti-inflammatory effect of ASD and the underlying mechanisms from the perspective of DNA methylation.” has been changed to: “This study aimed to preliminarily explore the anti-inflammatory effect of ASD and the underlying mechanisms from the perspective of DNA methylation and inflammation-related pathways.” (line 20-22).

Reviewer #4 comments

Finally, the writing requires extensive editing and, in some cases, the word choice obscures the science.

Our response: Thank you for your suggestion. The article has been edited for proper English language, grammar, punctuation, spelling, and overall style by one or more of the highly qualified native English speaking editors at AJE (Editing Certificate P4SYWN7H_2A9B-63DA-0775-E741-3DF8).

References

  1. Luo, J.F., et al., Activation of Nrf2/HO-1 Pathway by Nardochinoid C Inhibits Inflammation and Oxidative Stress in Lipopolysaccharide-Stimulated Macrophages. Front Pharmacol, 2018. 9: p. 911.
  2. Kang, J.-K., Y.-C. Chung, and C.-G. Hyun, Anti-Inflammatory Effects of 6-Methylcoumarin in LPS-Stimulated RAW 264.7 Macrophages via Regulation of MAPK and NF-κB Signaling Pathways. Molecules, 2021. 26(17): p. 5351.
  3. Chiu, L.-C., et al., Diterpenoid Compounds Isolated from Chloranthus oldhamii Solms Exert Anti-Inflammatory Effects by Inhibiting the IKK/NF-κB Pathway. Molecules, 2021. 26(21): p. 6540.
  4. Gong, L.-l., et al., Anti-nociceptive and anti-inflammatory potentials of Akebia saponin D. European Journal of Pharmacology, 2019. 845: p. 85-90.
  5. Alkreathy, H.M. and A. Esmat, Lycorine Ameliorates Thioacetamide-Induced Hepatic Fibrosis in Rats: Emphasis on Antioxidant, Anti-Inflammatory, and STAT3 Inhibition Effects. Pharmaceuticals, 2022. 15(3): p. 369.
  6. Lai, S.-C., et al., DNMT3b/OCT4 expression confers sorafenib resistance and poor prognosis of hepatocellular carcinoma through IL-6/STAT3 regulation. Journal of Experimental & Clinical Cancer Research, 2019. 38(1): p. 1-18.
  7. Moreira Lopes, T.C., D.M. Mosser, and R. Gonçalves, Macrophage polarization in intestinal inflammation and gut homeostasis. Inflammation Research, 2020. 69(12): p. 1163-1172.
  8. Park, M.Y., et al., Scutellarein Inhibits LPS-Induced Inflammation through NF-κB/MAPKs Signaling Pathway in RAW264. 7 Cells. Molecules, 2022. 27(12): p. 3782.

Round 2

Reviewer 4 Report

The authors do a good job responding to critique and should be lauded for their efforts. That said, the failure to include the ASD-alone control is a fundamental flaw in research design. We are all taught to include all relevant controls in every experiment and this work is no exception. Not including the most basic control is an insurmountable issue in experimental design. The author's retort that the response is expected given the properties of ASD, however, is an improper to assumption of the outcome without data showing this property in their hands. Without this control, the study is simply incomplete.